



# Evidence-based requirements for perceptualising intercatchment groundwater flow in hydrological models

Louisa D. Oldham[1]*, Jim Freer[1,2,3], Gemma Coxon[1,2], Nicholas Howden[4], John P. Bloomfield[5] & Christopher Jackson[6]

[1]School of Geographical Sciences, University of Bristol, Bristol, BS8 1SS, United Kingdom
[2]Cabot Institute, University of Bristol, Bristol, BS5 9LT, United Kingdom
[3]Centre for Hydrology, University of Saskatchewan, Canmore, Alberta, T1W 3G1, Canada
[4]Department of Civil Engineering, University of Bristol, Bristol, BS8 1TR, United Kingdom
[5]British Geological Survey, Wallingford, OX10 8BB, United Kingdom
[6]British Geological Survey, Keyworth, NG12 5GG, United Kingdom.

*Correspondence to*: Louisa Oldham (louisa.oldham@bristol.ac.uk)

**Abstract.** Groundwater-dominated catchments are often critical for nationally-important water resources. Many conceptual rainfall-runoff models tend to degrade in their model performance in groundwater-dominated catchments as they are rarely designed to simulate spatial groundwater behaviours or interactions with surface waters. Intercatchment groundwater flow is one such neglected variable. Efforts have been made to incorporate this process into existing models, but there is a need for improving our perceptual models of groundwater-surface water interactions prior to any model modifications.

In this study, national meteorological, hydrological, hydrogeological, geological and artificial influence (characterising abstractions and return flows) datasets are used to develop a perceptual model of intercatchment groundwater flow (IGF) and how it varies spatially and temporally across the River Thames, United Kingdom (UK). We characterise the water balance, presence of gaining/losing river reaches and intra-annual dynamics in 80 subcatchments of the River Thames, taking advantage of its wealth of data, densely gauged river network, and geological variability.

We show the prevalence of non-conservative river reaches across the study area, with heterogeneity both between, and within, geological units giving rise to a complex distribution of recharge and discharge points along the river network. We identify where non-conservative reaches can be attributed to IGF, and where other processes (e.g. surface water abstractions) are the likely cause. Through analysis of recorded water balance data and hydrogeological perceptualisation, we conclude that outcrops of carbonate fractured aquifers (Chalk and Jurassic Limestone) show evidence of IGF both from headwater to downstream reaches, and out-of-catchment via spring lines. We found temporal as well as spatial variability across the study area, with more seasonality and variability in river catchments on Jurassic Limestone outcrops compared to Chalk and Lower Greensand outcrops. Our results demonstrate the need for local investigation and hydrogeological perceptualisation within regional analysis, which we show to be achievable given relatively simple geological interpretation and data requirements. We support the inclusion of IGF fluxes within existing models to enable calibration improvements in groundwater-dominated catchments, but with geologically-specific temporal and spatial characteristics, and (when perceptually appropriate) connectivity between catchments.



## 1 Introduction

Groundwater-dominated river catchments are often critical for nationally-important water resources (Hartmann et al., 2017; Yang et al., 2017), with groundwater itself accounting for, on average, one third of global human water consumption (IGRAC, 2020). However, conceptual rainfall-runoff models used for the simulation of catchment river flows are rarely designed to simulate spatial groundwater behaviours or interactions with surface waters (Wanders et al., 2011). Many conceptual catchment models applied to groundwater-dominated catchments tend to degrade in their model performance metrics when

simulating observed river flow (Le Moine et al., 2005; Pellicer-Martinez et al., 2015; McMillan et al., 2016; Coxon et al., 2019; Lane et al., 2019), having to resort to the selection of unrealistic parameters (Goswami and O'Connor, 2010) or factoring of climate inputs or catchment area (Le Moine et al., 2007) to achieve acceptable calibration results. To enable improvements of conceptual models to address this lack of groundwater process representation, groundwater and surface water processes need to be holistically reviewed and conceptualised (Barthel and Banzhaf, 2016).

### 45 1.1 Intercatchment groundwater flow

Under the Tothian framework of groundwater flow, local, intermediate and regional groundwater systems can be present contemporaneously (Toth, 1963). At the subcatchment scale and under natural conditions, surface water boundaries are topographical whilst groundwater boundaries are a complex manifestation of topographic lows, the spatial distribution of recharge, and subsurface hydraulic properties that depend on lithology. Under the appropriate geological conditions, there is,

therefore, the potential for precipitation falling in one surface water catchment to flow to an adjacent surface water catchment's streamflow channel via the groundwater flow system (intercatchment groundwater flow (IGF)), across the topographical surface water divide and bypassing the 'donor' stream's outlet gauge (Le Moine et al., 2005; Gascoin et al., 2009; Pellicer-Martinez and Martinez-Paz, 2014; Barthel and Banzhaf, 2016; Bouaziz et al., 2018; Fan, 2019; Le Mesnil et al., 2020). IGF can result in a net loss or gain to a surface water catchment (termed a "non-conservative" catchment) and be an important term

in a catchment's water balance as both unmeasured, and often unaccounted for, inflow and outflow (Yeh et al., 1998; Bouaziz et al., 2018; Fan, 2019; Le Mesnil et al., 2020; Wagener et al., 2021). Yet the impact of IGF on catchments' water balances is an often neglected variable in conceptual hydrological modelling (Pellicer-Martinez and Martinez-Paz, 2014).

Studies have found IGF to be most prominent in headwater catchments (Bouaziz et al., 2018; Fan, 2019) of increasing slope

(Ameli et al., 2018), underlain by high-permeability aquifers (Le Moine et al., 2005; Schaller and Fan, 2009) and in drier climates (Schaller and Fan, 2009), but with geological spatial variability being the key control on the prevalence of IGF (Genereux et al., 2002; Schaller and Fan, 2009; Frisbee et al., 2016; Bouaziz et al., 2018; Le Mesnil et al., 2020). Correlations to IGF presence and magnitude have also been made to larger basin sizes (Schaller and Fan, 2009; Bouaziz et al., 2018), although this itself is an anthropogenic variable as it is dependent on the human placement of river gauging stations with often

no consideration of underlying hydrogeological flowpaths (Krabbenhoft et al., 2022). Water balance analysis (e.g. Schaller





and Fan (2009); Bouaziz et al. (2018)), geochemical analyses (e.g. Genereux et al. (2002); Frisbee et al. (2016)), geomorphic analysis (e.g. Frisbee et al. (2016)) and modelling methods (e.g. Zanon et al. (2014); Bouaziz et al. (2018)) have been used to spatially characterise and quantify IGF. The prevalence of intra-annual temporal variation in IGF has also been shown (Bouaziz et al., 2018). These studies have all emphasised the need for local-scale geological interpretation in the understanding of this
complex physical process (Fan, 2019).

## 1.2 Efforts to address the 'Watertight Substratum' challenge

In conceptual hydrological catchment models, groundwater tends to be represented via a secondary, deeper subsurface water store, with fluxes representing groundwater contribution from the saturated zone to river flows controlled by a baseflow parameter. Lumped conceptual models often calculate only one head of groundwater for an entire catchment (Wanders et al.,
2011). A closed water balance is routinely ensured, where a surface water catchment is set-up assuming topographical controls, underlain by a low permeability horizontal substratum and its fluxes self-contained (Pellicer-Martinez and Martinez-Paz, 2014; Barthel and Banzhaf, 2016) – termed the 'Watertight Substratum' approach (Le Moine et al., 2005).

A parameterised "loss function" has been incorporated by hydrologists into a number of conceptual rainfall-runoff models
such as PMS (Pellicer-Martinez et al., 2015), abcd (Pellicer-Martinez and Martinez-Paz, 2014), GR4J (Le Moine et al., 2005; Le Moine et al., 2007), SMAR (Goswami and O'Connor, 2010), HYPE (Lindstrom et al., 2010) and TopNet-GW (Yang et al., 2017). These flux functions can generate a permanent loss to the system in lumped models (e.g. GR4J (Le Moine et al., 2005)) and improve calibrations in groundwater dominated catchments. However, owing to an absence of connectivity between sub-catchments' deep groundwater stores, they do not represent the physical reality of IGF. In a semi-distributed hydrological
modelling framework, IGF between catchments can be modelled via pre-defined network linkages (e.g. abcd (Pellicer-Martinez and Martinez-Paz, 2014)) and temporal variability can be applied (e.g. TopNet-GW (Yang et al., 2017)). However, *a priori* perceptualisation of IGF spatial and/or temporal variability by hydrologists has been to-date solely based on observed accretion flow profiling (Yang et al., 2017) or loss/gain water balance output from a lumped catchment model (Pellicer-Martinez and Martinez-Paz, 2014), rather than hydrogeological perceptualisation.

## 1.3 Holistic surface water-groundwater perceptualisation

While the "loss function" approaches have improved model performance for surface hydrological predictions and incorporated a key 'missing' hydrological variable, the inclusion of these IGF losses in conceptual rainfall-runoff models seems to be rarely evaluated as to its realism and whether spatial patterns of losses relate to understanding of the geological controls in and between catchments. Editing groundwater processes within a model without a preceding perceptualisation of the groundwater
environment can be detrimental to model performance and lead to results at odds with the physical characteristics of a catchment (Hughes et al., 2015). Perceptualisation is the development of a perceptual model, whereby a system is described qualitatively (and, potentially, quantitatively) and is akin to a hydrogeologists' conceptual model (Wagener et al., 2021). (The



term "perceptual" is used by hydrologists in place of "conceptual" to avoid confusion with the type of hydrological mathematical model). Perceptual models are routinely developed by hydrogeologists (Wagener et al., 2021) prior to
groundwater modelling (e.g. Entec UK Ltd. (2008), Atkins (2010) and ESI Ltd. (2013)).

We believe that there is a need for greater emphasis on improving our holistic groundwater-surface water perceptualisations of surface water catchments, prior to any hydrological conceptual model edits. This would allow the modeller to make informed decisions and reduce the risk of a model edit being used as a proxy for input data errors or parameterisation limitations
(Goswami and O'Connor, 2010; Lane et al., 2019), whilst linking groundwater-surface water interaction theory back into hydrological analysis, as advocated by Fan (2019). This can be a challenging task as the scale of groundwater influences can be mismatched to the local hydrology, hydrogeological evidence available, and involve complex lithologies. A 'high-level' hydrogeological analysis is here advocated, focussed on the needs of the surface water modeller.

This paper presents a perceptualisation of IGF for the purposes of hydrological conceptual modelling. We use available national meteorological, hydrological, hydrogeological, geological and human influences datasets to characterise the variability of IGF in both time and space. We use the River Thames as a case study example, due to its wealth of data, densely gauged river network, geological variability and insights of previous geological and hydrogeological studies of the basin (e.g. Andrews (1962); Bloomfield et al. (2009); Bloomfield et al. (2011); Bricker and Bloomfield (2014); Mathers et al. (2014);
Environment Agency (2018a)). We characterise the water balance, gaining/losing river reaches, and the presence and seasonal dynamics of IGF, to better understand these processes on a regional (aquifer) and local (river reach) scale. We then discuss the application of fluxes to models to account for IGF processes, how they might best be applied given the characterisations we have developed, and the challenges and limitations associated with the process of IGF perceptualisation.

## 2 Study Area

The River Thames at Kingston (hereafter referred to as the Thames catchment), flows in a predominantly south-easterly direction (Fig. 1) and covers an area of 9,948 km$^2$, containing over 100 operational river gauging stations (Marsh and Hannaford, 2008). There is a diverse range of topography and land use, being predominantly rural in the west and increasingly urbanised in the east. The river's source is on the higher ground of the Jurassic Limestone Cotswold Hills in the west, flows down the Upper Thames Valley, through the Chalk escarpment of the Chilterns and Marlborough & Berkshire Downs, and
across the Lower Thames Valley (Fig. 1(a)). Numerous tributaries join the main river, including the Coln, Kennet, Colne and Mole, with the exception of the Chalk outcrop where there is limited surface water drainage (Fig. 1(b)). Catchments with the highest baseflow indices (BFIs) are predominantly found on the areas of higher ground where there are outcrops of Jurassic Limestone and Chalk (Fig. 1(c) and (d)). BFIs quantify the ratio of baseflow to total river flow, with geology being their primary control (Bloomfield et al., 2009). There is significant hydrological variability in terms of drainage network, gauged



catchment size (the smallest being only 2 km²) and hydrograph characteristics (BFIs ranging from 0.17 to 0.98 and mean flow from 0.01 to 78 $m^3s^{-1}$ (Marsh and Hannaford, 2008)). Water fluxes across the Thames catchment are substantially modified by human activities including surface water and groundwater abstractions, returns from sewage treatment works and reservoir operations (Environment Agency, 2018b; Bloomfield et al., 2021). Surface water abstractions for public water supply alone can exceed 50 $m^3s^{-1}$ in the Lower Thames (Harvey and Marsh, 2012).




**Figure 1: Maps of the Thames at Kingston catchment showing the River Thames at Kingston main river flowing west to east, in relation to (a) the topography and key geographical locations, (b) the river gauging station network and key catchments referred to later in the text, (c) catchment baseflow index (National River Flow Archive (Marsh and Hannaford, 2008) and (d) main bedrock lithology (after British Geological Survey (2016)).**

## 3 Methods

### 3.1 Data

We use the surface water topographical boundaries delineated by the 102 operational river gauging station network as our starting point, owing to their use as the delineation basis of surface water catchment models. In this study we refer to a river 'reach' as the catchment area between river gauging stations. The analysis undertaken in this study is developed at the river reach scale rather than at the sub-catchment scale, as similarly undertaken by Le Mesnil et al. (2020) in their study of the impact of IGF on the water balance of karst catchments in France. This approach has been adopted to avoid increasing averaging or 'smoothing' of further downstream metrics, therefore masking small-scale local variation. We collated hydro-meteorological, hydrogeological, geological and human influence data to in order to undertake the water balance analysis detailed in Sect. 3.2.

### 3.1.1 Hydro-meteorological timeseries

To characterise the water balance for each reach, daily precipitation, potential evapotranspiration (PET) and discharge for a 21-year period from 01 January 1994 to 31 December 2014 were obtained. Daily rainfall was sourced from the UK Centre for Ecology & Hydrology's (CEH) Gridded Estimates of Areal Rainfall (CEH-GEAR) dataset, a 1 km$^2$ gridded product that covers the whole of Great Britain and originates from the Meteorological Office's rain gauge observations (Tanguy et al., 2019). Estimates of daily PET were taken from CEH's Climate hydrology and ecology research support system PET dataset (CHESS-PE), which uses the Penman-Monteith equation and is also provided at a 1 km$^2$ gridded scale for Great Britain (Robinson et al., 2016). Reach average rainfall and PET timeseries were calculated by averaging values of all grid squares that lied within the reach topographic boundaries described above. Observed daily river flow data were obtained from the National River Flow Archive (NRFA) for all available gauges within the Thames catchment. Additionally, a daily timeseries of actual evapotranspiration (AET) was calculated by the Thornthwaite (1948) water budget method, based on the reach CHESS-PE potential evapotranspiration series and parameterised with average soil root depths from the work of Lane et al. (2021).

Daily timeseries of concurrent precipitation, AET and discharge were generated, with a day removed from the analysis if it did not contain all three of these variables of data. When calculating a monthly timeseries, up to five days of missing discharge data per month were allowed, provided the cumulative rainfall totals on the corresponding days were no greater than 10% of the total monthly rainfall volume. The missing days were then infilled with the monthly average of that specific month. Average January to December monthly profiles were then calculated, provided there was a minimum of 14 separate months (e.g.





Januarys) of data in the timeseries, out of the maximum of 21 in the time period of interest. Unfortunately, this meant that
170    some reaches of key geological interest (e.g. the upper River Lambourn reaches in the River Kennet catchment on the
Marlborough & Berkshire Downs Chalk (ref. Fig. 1(a) and (b)) were not able to be included in the analysis due to their low
data availability; however, a balance had to be made to ensure enough data for robust annual and inter-annual flow calculations.
80 of the original 102 reaches remained for use in the following analysis.

### 3.1.2 Groundwater level timeseries

Groundwater level data were used to confirm groundwater flow directions and investigate the degree of similarity between
temporal variations of surface water flows and groundwater levels. Raw data from 1,634 boreholes covering the Thames at
Kingston catchment were provided by the Environment Agency, quality assured for units, trends and outliers, and processed
into normalised monthly average timeseries, whereby the minimum groundwater level in an entire timeseries was subtracted
from all the datapoints in that same timeseries. Rather than using (often sparse or of variable quality) construction metadata,
it has been assumed that a borehole's data reflects the productive aquifer on which it is located. Only boreholes located in
reaches with a greater than 70% coverage of aquifer outcrop were reviewed, as it is on outcrops that groundwater (usually)
exerts the most significant influence on surface hydrology. For consistency, the data availability constraints applied during the
development of the monthly average groundwater timeseries were similar to the surface water flow series, ensuring there were
a minimum of 14 monthly average datapoints for each calendar month in the 21-year timeseries. This left 151 boreholes
remaining in the analysis (the locations of which are shown in Fig. 3(a)). The geographical location of a well within a reach in
relation to a river or interfluve etc. was not used as a selection factor, in the interest of keeping as many boreholes as possible
in the analysis. It is worth noting that many boreholes have historically only been sampled at a monthly timestep. 31%, 21%
and 48% of the boreholes had weekly, fortnightly and monthly data respectively. Sensitivity testing was therefore also
undertaken on the within-month temporal variability of groundwater levels and there was minimal effect.

### 190    3.1.3 Human Influences

Abstractions and discharges can have significant impacts on the evaluation of water balance reach scale calculations.
Consequently, such artificial influences should be included in water balance analyses to ensure that we do not over or under-
estimate a reach's water balance characteristics and bias our understanding of IGFs. Surface water abstractions and discharges
data were obtained from the CAMELS-GB dataset, which compiled data and information from the Environment Agency
(Coxon et al., 2020), and then processed it into reach totals based on the geographic location of the abstraction/discharge. The
accumulated abstraction/discharge impacts along a river were calculated and assigned to each reach. There are a numerous
uncertainties associated with naturalisation methodologies and resulting data, which are discussed in more detail in Sect. 3.2.1
and 6.3.



### 3.1.4 Superficial and bedrock geology

The British Geological Survey's 50k superficial and bedrock geology maps (British Geological Survey, 2016) were used to characterise a reach's underlying hydrogeology, by calculating the percentage of aquifer outcrop coverage within each reach. From this, a classification of four hydrogeological typologies was defined: Chalk, Jurassic Limestone, Lower Greensand, and non-aquifer (i.e. no aquifer outcrop). A limiting factor of 70% of the total reach area was assigned as an indicator of reach coverage. Dominant lithology has been used previously by Le Moine et al. (2005) in their review of geological variability in

relation to the presence of IGF in France.

### 3.2 Analysis

### 3.2.1 Spatial analysis

Firstly, we reviewed water balance data at a reach scale to quantify the long-term system characteristics of a catchment in relation to its core geological characteristics. The relationship between long term average precipitation, actual evaporation and

river discharge can be used to characterise the water balance of a reach. This long-term water balance (or imbalance) is an important assessment to conduct prior to modelling (Goswami and O'Connor, 2010) and can help to identify catchments with IGF (Genereux et al., 2002; Bouaziz et al., 2018; Le Mesnil et al., 2020) and/or permanent losses to deep groundwater systems as measurable data on these variables is rarely available.

Following the "watertight catchment" premise which most conceptual rainfall-runoff models follow, assuming a zero-flow

condition over the whole boundary, change in storage in the reach over the long term (several years) can be assumed to be negligable (Bouaziz et al., 2018) and the water balance would be presented as:

$$\frac{dS}{dt} = P - AET - Q \tag{1}$$

$$Q = Q_{ds} - Q_{us} \tag{2}$$

where $dS/dt$ = the change in reach storage over time, $P$ = reach areal precipitation, $AET$ = reach areal actual

evapotranspiration, $Q$ = reach river outflow, $Q_{ds}$= reach downstream recorded river flow, $Q_{us}$ = reach upstream recorded river flow (if present), and units are in millimetres per year. The difference between the upstream gauge and the downstream gauge ($Q$) gives the additional surface water flow contribution from within that reach only. A positive residual result from Eq. (1) can be indicative of a 'loss' of water to the reach, where river flow leaving the reach is less than expected from the meteorological data, and conversely a negative residual result a 'gain' of water to the reach.


Partial-naturalisation of river flows was undertaken following the 'naturalisation by decomposition' method (Environment Agency, 2001), whereby surface water abstractions and discharges are applied to the recorded river flow record but groundwater abstractions are excluded from the naturalisation. Whilst this may be an acceptable assumption for many rivers reaches, even on aquifer outcrops, it certainly will not be true for all (Wendt et al., 2020). It was felt, however, that a more





detailed analytical approach involving groundwater abstractions was outside the scope of this study owing to the complexity
of spatially allocating groundwater abstraction impacts to particular river reaches (see Sect. 6.3 for further discussion). Under
the assumption that all abstracted water is lost to the system, naturalising the recorded river flow record would result in the
water balance equation becoming:

$$\frac{dS}{dt} = P - AET - Qnat \tag{3}$$

$Q_{nat} = (Q_{ds} - Q_{us}) + A - D$ (4)

where $Q_{nat}$ = partially-naturalised reach river outflow, $A$ = reach surface water abstractions and $D$ = reach surface water
recorded discharges and units are in millimetres per year.

Whilst the annual average water balance  provides a clear losing/gaining assessment of a reach, the non-dimensional
representation of reach water balance metrics in relation to the Water Limit and Energy Limit considers there to be a wider
range of feasible physical characteristics (Le Moine et al., 2007; Andreassian and Perrin, 2012; Bouaziz et al., 2018). The
Water Limit represents the point above which a reach's outflow would be greater than its inflow (i.e. *Q/P>1*); a therefore
"gaining" reach. The Energy Limit conversely represents the point below which a reach must be "losing" water as runoff
deficits would be exceeding the maximum total PET. This analysis can therefore be used to identify reaches plotting beyond
the 'natural' range in terms of their water balance.

### 3.2.2 Temporal analysis

Monthly analysis of the variation in reach responses between the four hydrogeological typologies was undertaken to investigate
how seasonal responses vary between different groundwater systems. Reviewing surface water seasonality against local
groundwater level seasonality and meteorological variables enables a comparison of the influence of subsurface processes on
river flow temporal characteristics. Average October to March monthly profiles were developed for each reach in each geology,
covering effective rainfall (precipitation less actual evapotranspiration), groundwater level, discharge and water balance. A
comparison between the seasonal highs and lows of the different variables was undertaken.

### 3.2.3 Hydrogeological analysis

Local scale analysis and interpretation of meteorological, hydrological and hydrogeological data allows the hydrologist to
refine a perceptual model into a more subcatchment-focussed understanding. Whilst the literature states that a reach's water
balance (Bouaziz et al., 2018), its location upstream (Fan, 2019) and the presence of permeable substratum (Le Moine et al.,
2005) can all assist in the identification of reaches with IGF, we propose developing this further via hydrogeological analysis,
whereby we then look at the location of non-conservative reaches' boundaries in comparison to aquifer outcrop boundaries
(Pellicer-Martinez and Martinez-Paz, 2014), aquifer properties, the local groundwater flow direction (Toth, 1963) and the




presence of springlines (Frisbee et al., 2016) to develop a more detailed reach-specific understanding of the groundwater environment and provide additional evidence of IGF. The perceptual 'roadmap' (Fig. 2) can assist with the identification and explanation of both expected and unusual water balance reach characteristics. A final perceptual model of the Thames catchment is then presented in Sect. 6.1.

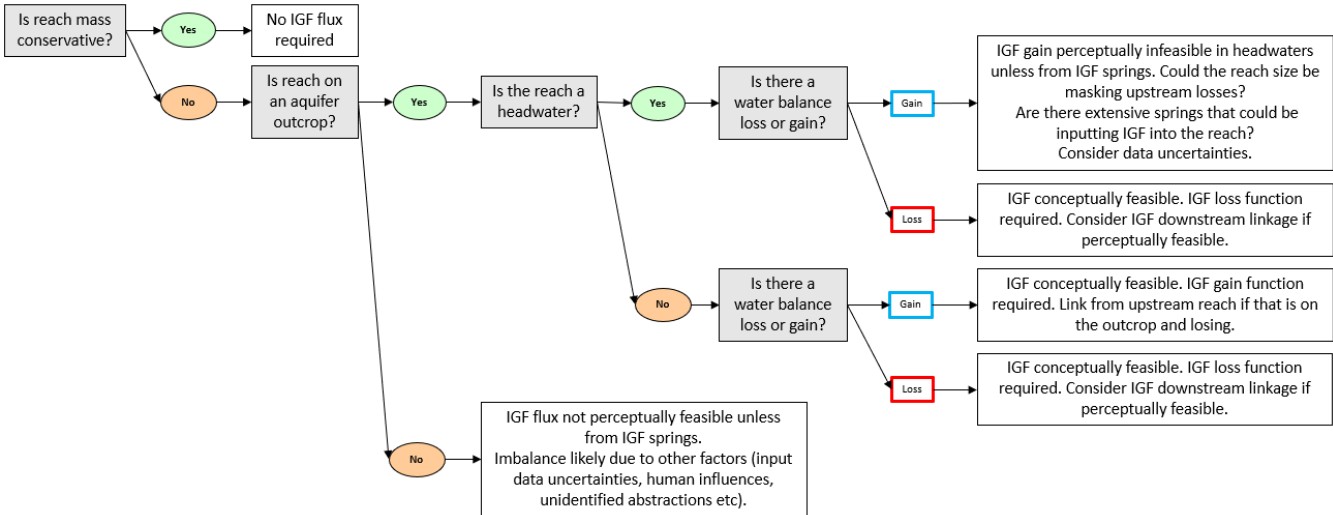


**Figure 2: Perceptual 'roadmap' for high-level initial identification of river reaches with possible intercatchment groundwater flow, from a physical process-based assessment. Note that more than one process may be at work in a single reach, e.g. both incoming IGF from upstream river reaches and outgoing IGF to downstream river reaches.**

## 4 Hydrogeological review

To fully understand IGF across the River Thames Basin we need to first undertake a local analysis of the geology and hydrogeology. This information in imperative in supporting any conclusions as to the presence of IGF in a study area.

The geology of the Thames is highly heterogenous. The geological strata at the surface become progressively younger from north-west to south-east across the catchment (Fig. 3(c)) and give rise to 'scarp and vale' topography (Lane et al. 2008).
Superficial deposits are not continuous (although they may be hydrogeologically significant at the river reach scale (Bloomfield et al., 2011; Bricker and Bloomfield, 2014). There are three principal aquifers within the Thames Basin: the Chalk, the Jurassic oolitic limestones, and the Lower Greensand (Fig. 3). The groundwater in these aquifers is separated by lower permeability geological strata (Fig. 3(c)) and so, for the purposes of perceptual modelling, can be considered to be disconnected, complicating the regional groundwater flow framework of Toth (1963) as there will not be a Thames catchment-wide
groundwater flow system. There will, however, be aquifer-wide groundwater flow systems which may link surface water catchments that are situated on the same aquifer outcrop via IGF pathways. The groundwater level data in Fig. 3(a) and contours in Fig. 3(b) show groundwater flow at the aquifer scale is generally towards the south-east, with the exception of the





Lower Greensand, in which flow is towards the north-west (Fig. 3(b)). The aquifer properties of the three aquifer units are summarised in Table 1.

## 4.1 Jurassic Limestone

Groundwater is present in both the lower Inferior Oolite and upper Great Oolite Jurassic Limestones of the Cotswold Hills, which are in direct hydraulic contact in some areas and separated by the 20-30 m thick mudstone aquiclude of the Fullers Earth Formation in others (Bricker et al., 2014). The Jurassic Limestones have low matrix porosities, being well-cemented, and low intergranular permeability, but are highly fractured, faulted and fissured (Allen et al., 1997). Consequently, the aquifer has a high transmissivity and low storage coefficients (Table 1), with groundwater flow being predominantly along preferential flow paths, and groundwater levels responding rapidly to rainfall (Newmann et al., 2003). The aquifer extends across a number of river subcatchments (see Fig. 1c for distribution of subcatchments). Given the high transmissivity and presence of numerous springs across the outcrop, in particular in the upper reaches (Fig. 3(b)), perceptually there is the potential for IGF from one subcatchment to another. The location of many springs just outside of the study area, to the northwest of the topographical surface water watershed boundary, could be suggestive of IGF out of the Kingston catchment.

## 4.2 The Chalk

The Chalk outcrop covers 29% of the study area and contains over 20 river subcatchments (see Fig. 1c). The Chalk aquifer is the most important groundwater resource in Great Britain (Allen et al., 2009) and makes up over 50% of total groundwater abstraction (Shand et al., 2003). It is a fine-grained, microporous limestone, which has a high matrix porosity, but low matrix conductivity because of the small pore sizes (Butler et al., 2012). It is the Chalk's extensive fracturing that gives it its high transmissivity (Table 1) and through which saturated flow occurs. The fracture network also provides its drainable porosity; typically specific yield is the range 0.5-2%. Dissolution of the Chalk within the zone of water table fluctuation, and periglacial weathering, which has preferentially occurred in valleys, has enhanced the fracture network and given rise to significant vertical and horizontal heterogeneity. The extent of groundwater-surface water interaction across this geological unit will therefore be highly variable, dependent on very localised physical conditions of the Chalk. Bulk hydraulic conductivity generally decreases with depth, and transmissivity reduces away from river valleys (Allen et al., 1997). There is a clear sparsity of perennial rivers on the Chalk outcrop (Fig. 1(b)) indicating a predominance of groundwater flow mechanisms (Adams, 2008). Given these physical attributes, the groundwater flow network across the aquifer will be having a significant impact on the surface water hydrology. Springlines are evident along the outcrop boundaries, particularly along the north-western escarpment of the Chalk (Fig. 3(b)), where the groundwater contours in Fig. 3b show localised northerly groundwater flow off the Chalk outcrop. This is likely causing IGF into the Upper Thames Valley.




### 4.3 Lower Greensand

The Lower Greensand Group comprises two main hydrogeological units – the Hythe Formation and the uppermost Folkestone
Formation, which, being separated by an aquitard layer, can be considered two separate aquifers (Allen et al., 1997).
315 Intergranular flow is the primary flow mechanism, along with fracture flow (Shand et al., 2003a). Due to the high storage
(Table 1), there is limited seasonal variation in groundwater head (Bloomfield et al., 2011). There is a greater density of
drainage channels on the Lower Greensand than the Chalk or Jurassic Limestone outcrops (Table 1 and Fig. 1(b)) and a
concentration of springs in the upper reaches (Fig. 3(b)). Owing to the lower river incision and transmissivity (Table 1), the
impact of groundwater-surface water interaction on the river flows in subcatchments located on this outcrop may be less than
320 on the Jurassic Limestone and Chalk aquifer outcrops.

**Figure 3: Hydrogeological features and characteristics of the Thames at Kingston catchment showing (a) average groundwater levels
and the location of the geological cross section in (c), (b) the location of springs (digitised from Ordnance Survey mapping), general
'median' aquifer groundwater level contours (after Atkins (2003), Atkins (2007), Entec UK Ltd. (2008), Atkins (2010) and ESI Ltd.
325 (2013)) and flow direction arrows in relation to aquifer outcrop areas and (c) a geological cross section showing the three main
aquifers, simplified lithology and general groundwater flow direction.**





**Table 1: Range of hydraulic aquifer properties (where available) of the main aquifers in the Thames at Kingston catchment. After Allen et al. (1997) and Bloomfield et al. (2011). It should be acknowledged that the storage values reported below may relate to both confined (elastic) storage coefficients and specific yield values, depending on the approach taken to analyse the pumping tests on which they are derived. This may be particularly likely in the Chalk.**

| Hydraulic property | | Jurassic Limestone | Chalk | Lower Greensand |
|---|---|---|---|---|
| **Aquifer lithology** | | Fractured aquifer | Fractured aquifer | Intergranular aquifer |
| **Log transmissivity (m²d⁻¹)** | | | | |
| | **Mean** | 2515 | 1766 | 430 |
| | **Median** | 790 | 855 | 275 |
| | **Interquartile range** | 200-1600 | 243-1800 | 188-528 |
| **Storage** | | | | |
| | **Mean** | 0.013 | 0.015 | 0.005 |
| | **Median** | 0.0002 | 0.01 | 0.0002 |
| | **Interquartile range** | 0.0002-0.0005 | 0.003-0.02 | 0.0001-0.0005 |
| **River incision (m)** | | 43.07 | 55.01 | 24.8 |
| **Drainage density (km/km²)** | | 0.56 | 0.54 | 0.95 |

# 5 Results

## 5.1 The spatial variation of annual water balance metrics

Figure 4, 5 and 6 present the spatial variation in average annual water balance metrics (1994-2014) across the Thames catchment. The results are discussed with reference to the hydrogeological review undertaken in the previous section.

There is minimal difference between the non-naturalised and the naturalised results (Fig. 4, Fig. 5 and Fig. 6), with a non-conservative water balance of >100 mm yr⁻¹ being 'corrected' by naturalising the discharge series in only three of the 80 reaches. The three lowest main river reaches show particularly large naturalised water balance losses (>1000 mm yr⁻¹) (Fig. 5(b)). In a catchment as heavily artificially influenced by abstractions and discharges as the Thames at Kingston (Environment Agency, 2018b) this raises questions as to the completeness of the naturalisation data and/or the effectiveness of the employed naturalisation methodology. This is discussed further in Sect. 6.3. For the purposes of the current analysis, focus is given to the naturalised results only, owing to their similarity to the non-naturalised results.

The naturalised results show variability in annual water balance between the main hydrogeologies – Non-aquifer areas, Jurassic Limestone outcrop, Chalk outcrop and Lower Greensand outcrop (Fig. 4(b)). With the exception of the three lowest main river Thames reaches near Kingston, annual reach losses are only observed on aquifer outcrop areas (Fig. 5(b)), where the recorded




river flow is less than expected given the meteorological variables. 60% of the greatest water balance losses (>100 mm yr$^{-1}$)

observed in the Thames catchment are on the Chalk outcrop, where there is an average reach loss of 187 mm yr$^{-1}$. The Lower
Greensand reaches show smaller water balance losses on average (51 mm yr$^{-1}$) (Fig. 4(b)), with only one of the four reaches
exhibiting a loss of more than 100 mm yr$^{-1}$ (Fig. 5(b)). The Jurassic Limestone reaches gain water on average (average water
balance of -156 mm yr$^{-1}$) although there are some reaches with large losses (Fig. 4(b)). Again with the exception of the three
lowest main river Thames reaches, non-aquifer reaches are gaining water at the annual scale on average (-60 mm yr$^{-1}$) (Fig.

4(b)). The results highlight significant variability within, not just between, the aquifer outcrop geologies. The Jurassic
Limestone reaches show the greatest range of naturalised annual average water balances (interquartile range of 228 mm yr$^{-1}$
compared to a range of 150 mm yr$^{-1}$ on the Chalk and 112 mm yr$^{-1}$ on the Lower Greensand reaches).

Reaches with negative annual water balances tend to be seen downstream of reaches with positive water balances (Fig. 5(b)).

The Chalk reaches show a pattern of headwater catchments consistently 'losing' (plotting below the Energy Limit) and non-
headwater catchments 'gaining' water at the annual average scale (Fig. 5b and Fig. 6g). For example, the Rivers Kennet and
Colne on the Chalk (ref. Fig. 1(b)) have annual average water balance losses in their headwater reaches (622 mm yr$^{-1}$ annual
loss in total in the Kennet headwater reaches) and gains further downstream (913 mm yr$^{-1}$ annual gain in total in the Kennet
lower reaches) (Fig. 5(b)). This is at the southerly limit of the Chalk aquifer outcrop, where it becomes overlain by the less

permeable Clay material of the Upper/Lower Thames Valleys (illustrated on Fig. 1(c) and Fig. 3). This can also be seen in the
River Coln on the Jurassic Limestone (ref. Fig. 1(b) and Fig. 5(b)).

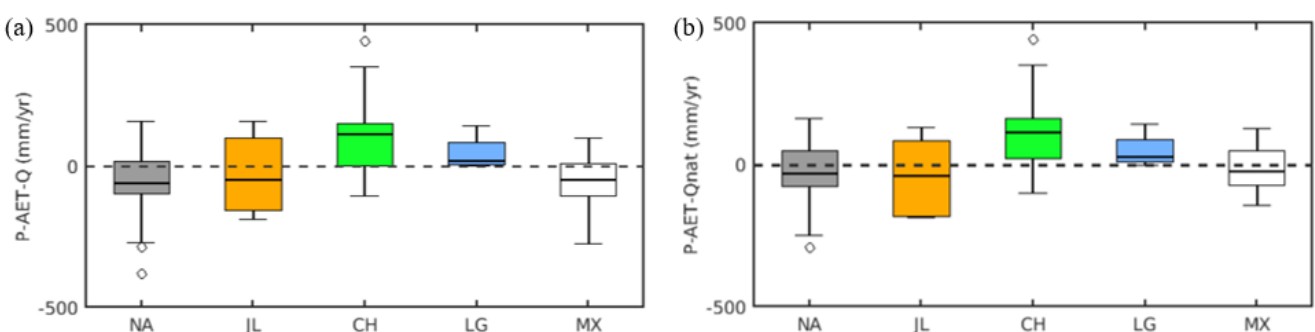

**Figure 4: Distribution of annual average reach water balance metrics for each lithology for the Thames at Kingston reaches from 1994-2014 inclusive, showing unaccounted for annual water volume from precipitation after subtraction of actual**

**evapotranspiration and (a) non-naturalised/(b) naturalised river flow (in millimetres per year). A positive water balance residual indicates a "losing" reach and a negative water balance residual a "gaining" reach at the annual time scale. The boxes show the interquartile interval, within which lies 50% of the data, and the horizontal line the median value. The whiskers show the minimum and maximum values excluding outliers. The plots are focussed in to +-500 mm yr$^{-1}$, thereby excluding some wider outliers. Reaches have been categorised based on >70% catchment geological coverage. CH = Chalk (n=23), JL = Jurassic Limestone (n=11), LG =**

**Lower Greensand (n=4), NA = Non-aquifer (n=28) and MX = Mixed (n=14).**





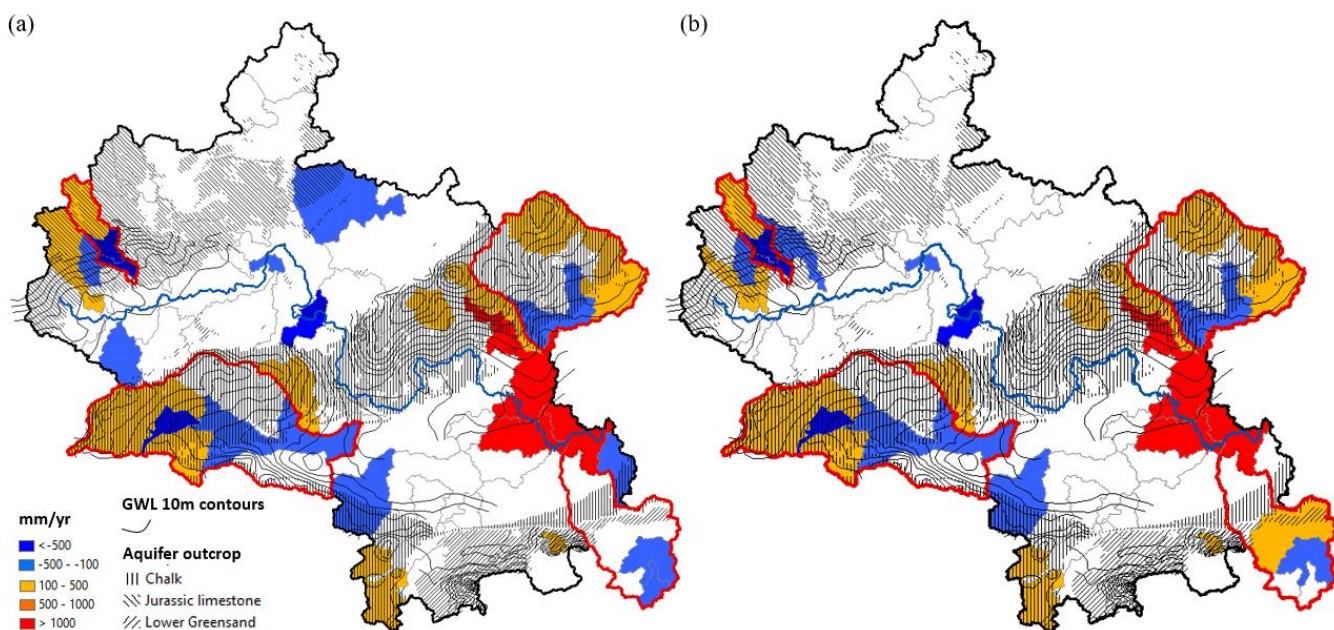

**Figure 5: Annual average reach water balance metrics for the Thames at Kingston from 1994-2014 inclusive, showing unaccounted for annual water volume from precipitation after subtraction of actual evaporation and (a) non-naturalised and (b) naturalised river flow (in millimetres per year), in relation to aquifer outcrop areas and median groundwater level contours for the same time period. A reach where the water balance residual is within 100 mm of balanced is considered to be conservative, to nominally account for data uncertainties. Catchments referred to in the text are outlined in red.**

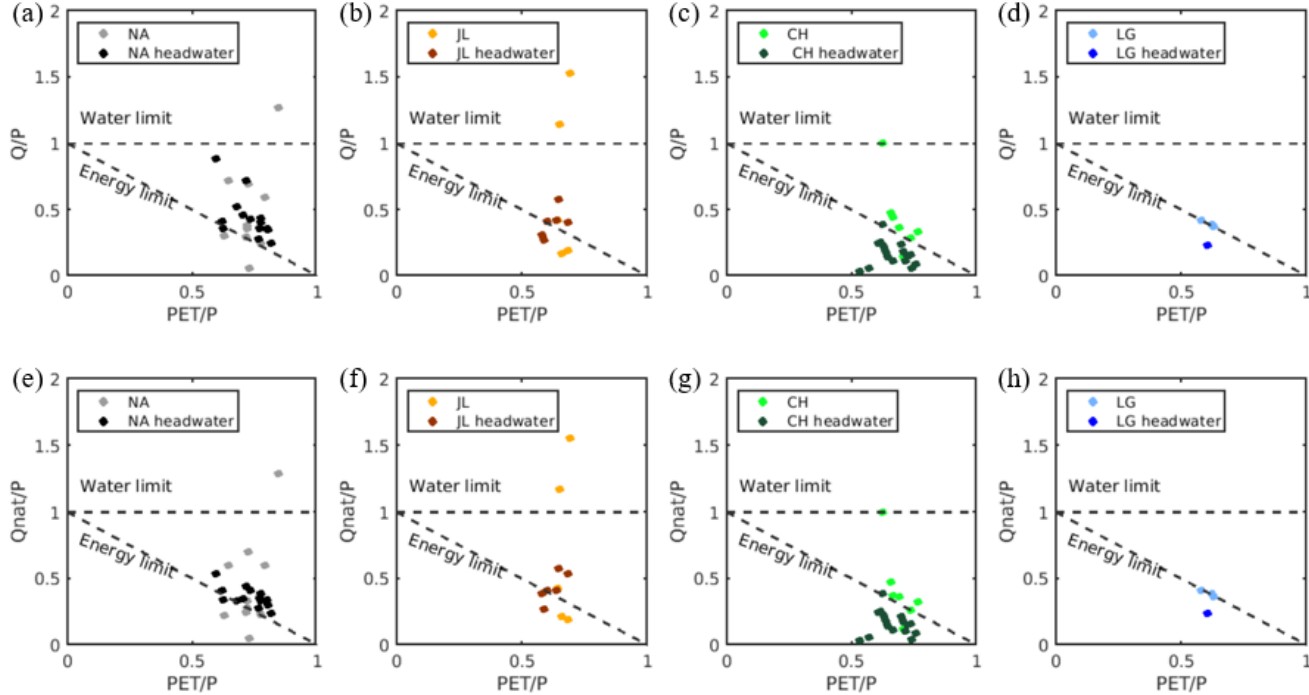






**Figure 6: Annual average reach water balance metrics for the Thames at Kingston reaches from 1994-2014 inclusive, showing dimensionless reach runoff coefficient (river flow/precipitation) and dryness index (potential evapotranspiration/precipitation) in relation to the Water Limit, Energy Limit and their headwater (i.e. no upstream gauge) or 'non-headwater' location along the river, under non-naturalised (a) to d) and naturalised (e) to (h) conditions. Reach categorisations are based on >70% catchment geological coverage of NA = Non-aquifer (n=28), JL = Jurassic Limestone (n=11), CH = Chalk (n=23) and LG = Lower Greensand (n=4). The results from the three Lower Thames reaches are not shown on figures (a) and (e) as they have negative reach runoff coefficient results.**

## 5.2 The temporal variation of annual water balance metrics

There is little difference in the seasonality of effective rainfall between the different geologies (Fig. 7(a)-(d)) across the Thames Basin but the intra-annual temporal variability of reach average groundwater levels (Fig. 7(e)-(g)) does vary, illustrating the differences between the physical aquifer properties of the three hydrogeological units.

The Lower Greensand annual average groundwater hydrographs are particularly muted (Fig. 7(g)), with minimal monthly variation in groundwater level (only a 0.6 m seasonal average range, in comparison to the 2.9 and 3.9 m range in the Jurassic Limestone and Chalk respectively – Table 2) and a particularly flat seasonal river flow profile (Fig. 7(k)) (Shand et al.; 2003a).

The Chalk reaches also show flat seasonal runoff coefficient (*Q/P*) profiles (Fig. 7(j) and Table 2), with the exception of one reach – the mid-reach on the River Kennet, which gains significant flow (see Fig. 5(b)). The Chalk reaches show a delayed onset of winter higher groundwater levels (March) in comparison to reaches on the Jurassic Limestone (January). The lag between rainfall and river flow is the longest of the geologies (Table 2). The signal of precipitation in the river flow record on the Chalk is, on average, a 4 month lag between average annual peak precipitation and average annual peak discharge (Table 2).

Rivers on the Jurassic Limestone exhibit the greatest monthly variability (Fig. 7(i) and Table 2), reflecting the groundwater trends within that geological unit (Fig. 7(e)). River flow seasonality on the Jurassic Limestone mirrors the highly responsive groundwater levels.





**Figure 7: Seasonal patterns of (a)-(d) effective rainfall (*P-AET*), (e)-(g) normalised groundwater levels, (h)-(k) naturalised reach runoff coefficient (*Qnat/P*) and (l)-(o) water balance (*P-AET-Qnat*) for non-aquifer, Jurassic Limestone, Chalk and Lower Greensand reaches. Reaches categorised based on a minimum of 70% geological aquifer outcrop coverage at the surface.**

**Table 2: Seasonality and lags of the 50th percentiles of precipitation, groundwater level and river flow. Reaches categorised based on a minimum of 70% geological aquifer outcrop coverage at the surface.**

| Water balance variable | Non-Aquifer | Jurassic Limestone | Chalk | Lower Greensand |
|---|---|---|---|---|
| Peak precipitation | November | October | November | November |
| Minimum precipitation | March | March | March | June |
| Peak groundwater level | - | January | March | April |
| Minimum groundwater level | - | September | November | October |
| Peak river flow | January | January | March | January |
| Minimum river flow | September | September | September | September |
| Seasonal range in river flow | 41 mm m$^{-1}$ | 44.5 mm m$^{-1}$ | 11.8 mm m$^{-1}$ | 36.0 mm m$^{-1}$ |
| Seasonal range in groundwater level | N/A | 2.9 m | 3.9 m | 0.6 m |



| | | | | |
|---|---|---|---|---|
| **Lag between peak precipitation and peak river flow** | 2.5 months | 3 months | 4 months | 2 months |
| **Lag between peak precipitation and peak groundwater level** | N/A | 3 months | 4 months | 5 months |
| **Lag between peak groundwater level and peak river flow** | N/A | None | None | 3 months |

## 6 Discussion

### 6.1 The Thames Perceptual Model


The perceptualisation process (illustrated in Fig. 2) enables us to recognise where IGF processes may be occurring. This highlights the need for a degree of local investigation and hydrogeological perceptualisation (Le Moine et al., 2005) within the regional analysis. Drawing on the findings from the analysis in the previous section, coupled with the hydrogeological review in Sect. 4, a groundwater-surface water regional perceptual model of the Thames catchment (from the perspective of a surface

water modeller) is provided below, and summarised in Fig. 8.





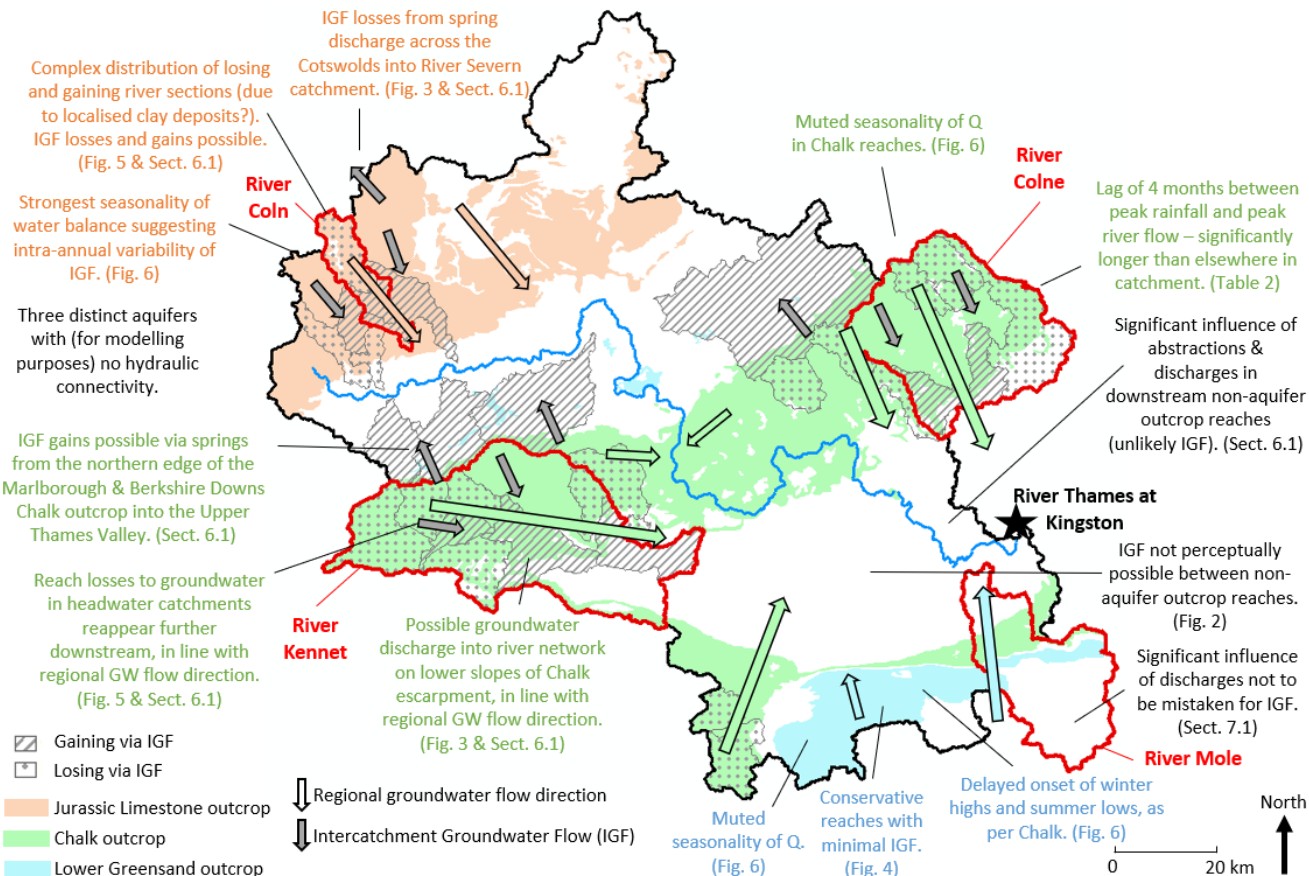

**Figure 8: Perceptual model of the Thames at Kingston catchment, including key modelling-related groundwater-surface water interaction features and characteristics. The Jurassic Limestone, Chalk and Lower Greensand aquifers are (for modelling purposes) hydraulically disconnected.**

Through analysis of recorded hydrometeorological water balance data we have shown the prevalence of non-conservative reaches across the Thames at Kingston catchment, both gaining and losing. This has also found by numerous authors in their analyses of annual water balance metrics, for example in France (e.g. Le Moine et al. (2007); Andreassian and Perrin (2012); Bouaziz et al. (2018); Le Mesnil et al. (2020)) and the USA (e.g. Schaller and Fan (2009)).

The spatial variability of non-conservative reaches has shown heterogeneity both between, and within, geological units. The significant heterogeneity of the Chalk (Bloomfield, 1996; Upton and Jackson, 2011) gives rise to both recharge and discharge points along river courses and, potentially, within river reaches (Butler et al., 2012). In the Jurassic Limestone, faulting and spatial variability in the thickness of the Fuller's Earth Formation (Sect. 4.1) between the Inferior and Great Oolites, and the large number of consequential springs (Fig. 3(b)), are leading to a complex distribution of discharge points across the outcrop

(Newmann et al., 2003). Although the sample size of the Lower Greensand reaches is small (n=4), we found limited variability



in water balance results between the Lower Greensand reaches which is in line with the recorded transmissivities for that unit being both smaller and of less variability than those found in the other geologies (Allen et al., 1997).

Fan (2019) recommends using the 'hydrologic signals' of adjacent losing to gaining reaches and the presence of spring clusters
to help identify IGF. Given the high porosity and extensive fracturing of Chalk (Sect. 4.2) (Bloomfield 1996) and the observed loss-to-gains in flow (Fig. 5(b)) which coincide with the regional groundwater flow directions depicted from average groundwater level data and contours in Fig. 3, it can be inferred that the 'lost' water in the upper Chalk reaches is flowing into adjacent downstream reaches via the groundwater flow system. This is supported by the literature (Allen et al., 1997; Shand et al., 2003b). Examples can be seen in the Rivers Kennet and Misbourne (Fig. 5(b)) where water balance gains are observed
as a result of local connectivity of geological units and the presence of springs along the contact of the Chalk and the overlying clay/mudstone (Fig. 3(b)) (Upton and Jackson, 2011).

In the literature, much research has looked at IGF in karstic environments (e.g. the work in France by, amongst others, Bouaziz et al. (2018), Le Mesnil et al. (2020) and Le Mesnil et al. (2021)). The Chalk of the Thames catchment can be locally sub-
karstic (Bloomfield et al., 2009; Upton and Jackson, 2011), but fracture and fissure flow remain the primary groundwater flow processes in these areas (Allen et al., 1997). Whilst the regional groundwater flow direction in the Chalk is to the south-east, localised northerly groundwater flow will be discharging Chalk groundwater from headwater reaches of the River Kennet into the "Non-aquifer" reaches of the Upper Thames Valley (Upton and Jackson, 2011) (see Fig. 3(b) groundwater contours and Fig. 1(a)). Whilst not impacting the water balance beyond 100 mm yr$^{-1}$ (and so not showing in Fig. 5(b)), the reaches just to
the north of the Marlborough & Berkshire Downs Chalk do exhibit water balance gains of 70-80 mm yr$^{-1}$, which are likely attributable to northwards IGF (ref. the perceptualisation roadmap in Fig. 2).

Karstic flow has also been reported in the Jurassic Limestone (Allen et al., 1997). IGF will likely be occurring in this unit via fracture pathways (Bricker et al., 2014) and from the springs to the north-west (into the adjacent river catchment to the Thames,
the River Severn (ref. spring locations in Fig. 3(b)), see Fig. 8). This is supported by the literature (e.g. Newmann et al. (2003)). The relationship between the Inferior and Great Oolite units is complex and cross-catchment flow has been confirmed (Bricker et al., 2014). Our results mirror this, particularly in light of the spatially variable nature of the losses and gains observed at the annual water balance (Fig. 5(b)).

From the above, we can attribute 'large-scale' non-conservatism of reaches on the Chalk and Jurassic Limestone to IGF, but this is not evident on the Lower Greensand reaches, which are largely conservative. Additionally, and crucially, water balance imbalances on non-aquifer outcrop reaches should not be attributed to IGF (unless located adjacent to an aquifer reach and down gradient of local groundwater flow, e.g. the spring flow into the Upper Thames Valley discussed above) (as we have assumed in the perceptualisation roadmap in Fig. 2).




Geology influences lag timescales (Marchant and Bloomfield, 2018; Fowler et al., 2020), with river flow seasonality showing a high level of influence of catchment storage. Our results demonstrate lag times of four months for Chalk reaches, in line with other studies (Weedon et al., 2015). The longer seasonal lags observed on the Chalk in comparison to the Jurassic Limestone (Table 2) will be due to the lower transmissivity coupled with high storage of that unit (Table 1). Whilst rapid flow through fractures can occur (Upton and Jackson, 2011), unsaturated flow through the Chalk is predominantly via the matrix, resulting in recharge of the water table via the displacement mechanism (Ireson et al., 2006). It is this process that is being principally observed in the relatively muted and delayed seasonal groundwater level hydrographs and, consequently, the surface water hydrographs in Fig. 7(f) and (j).


The Jurassic Limestone groundwater levels exhibit the greatest seasonal variation of the geologies owing to the low storage of the aquifer but high transmissivity (Bricker et al., 2014) (Table 1), as higher groundwater levels in the winter months give rise to the highest transmissivities and greater groundwater flow, but under drier conditions the opposite would occur. The presence of rapid flow pathways to the heavily incised (Table 1) rivers on that outcrop, via fractures and faults, results in a strong seasonal control on the river flow annual profile (Newmann et al., 2003; Bricker et al., 2014) that is not observed on the Chalk

or Lower Greensand.

Temporal variation is not always seen within IGF. Both Yang et al. (2017) and Zanon et al. (2014) found a temporally steady groundwater model flux to be successful for model calibration (although this was not linked back to the hydrogeological processes controlling the flux). In our example study, the high specific yield of the Lower Greensand aquifer gives rise to

observed steady groundwater heads (Shand et al., 2003a; Bloomfield et al., 2011) (Fig. 7(g)) and, combined with the high connectivity to the river drainage network via diffuse groundwater flow (Bloomfield et al., 2011), this is replicated in the muted seasonality of the Lower Greensand reaches' river flow profiles (Fig. 7(k)).

This perceptual model is the key output of the perceptualisation approach advocated by this paper, incorporating both surface

water and groundwater processes via an evidence-based objective analysis at the local reach scale. It is this output that we advocate being used by surface water modellers to aid with hydro(geo)logically appropriate model set-up, development and calibration.

## 6.2 Are current modelling approaches fit for purpose and how might we change them?

Not including IGF as a model flux will result in many models overestimating river flows or actual evapotranspiration (Bouaziz

et al., 2018; Fan, 2019) in an effort to 'close' the water balance in groundwater-dominated catchments. It is well known that the 'watertight substratum' assumption on which so many of our conceptual catchment rainfall-runoff models are built is frequently unrealistic in terms of the actual physical processes occurring (Le Moine et al., 2005; Wanders et al., 2011; Bouaziz



et al., 2018; Fan, 2019) as many of the key assumptions used to simplify a complex catchment (i.e. that of a clear boundary condition at the watershed, and a lower no-flow boundary condition at bedrock) do not hold for permeable, groundwater dominated catchments. We have the opportunity to develop the physical realism of these models and improve their simulation but, crucially, in-line with available evidence. We need to advocate different modelling approaches, with flexibility a key principle. Allowing high resolution compartmentalisation (e.g. Hydrological Response Units) of different process representations would enable the modeller to address the heterogeneity between different geologies and, even, within geologies at the sub-reach scale.

The extent of non-conservative reaches across our study area supports the application of external subcatchment flux(es) ("loss functions") to represent IGF, as previously advocated by Le Moine et al. (2007), Pellicer-Martinez and Martinez-Paz (2014) and Yang et al. (2017). We have, however, shown IGF to vary in time and space as a function of hydrogeological control. IGF may not be perceptually justified in all sub-catchments and should therefore be supported by prior perceptualisation investigations to inform decisions on flux flow direction, connectivity, temporal variability and spatial location. As such, we would not advocate the use of uniformly parameterised loss functions, where they are applied across all catchments in the same way. Parameterisation of IGF fluxes could be based on lithology, as suggested by Le Moine et al. (2005), or groundwater data (depending on data availability).

Loss functions should be linked between catchments where possible and hydrogeologically feasible (as highlighted by Pellicer-Martinez and Martinez-Paz (2014)). Groundwater flow is a process that occurs between surface water catchments and should be represented as such to reduce the risk of a model using such a flux as a simple 'fudge factor' to improve calibration (as warned by Le Moine et al. (2005) and Goswami and O'Connor (2010)). Flow accretion can show a longitudinal train of gaining and losing reaches, as shown by our reach-based analysis (Fig. 5(b)). Considering such reaches as separate, independent entities with independent IGF fluxes would misrepresent the hydrological processes occurring and could give rise to unfeasible parameterisation during model calibration.

The intra-annual variability of groundwater processes can be significant, based on natural cycles of meteorological and surface water variables. In addition, groundwater levels may be seasonally affected by abstractions (e.g. for agriculture). Losses are unlikely to be temporally continuous and so should not be represented as a constant loss with no seasonal variation. A constant value of flux should only be applied when the evidence and perceptualisation support this.

### 6.3 What are the challenges/limitations in perceptualising?

The measurement of IGF fluxes is rarely feasible (Le Moine et al., 2007; Frisbee et al., 2016), although there are examples of field studies attempting this e.g. Kaser and Hunkeler (2016), Genereux et al. (2005). Table 3 summarises the challenges faced when trying to perceptualise IGF, linking hydrological and hydrogeological data and varying spatial scales, categorised into:



data challenges, methodological challenges and perceptual understanding challenges. Two key challenges (those of uncertainties and naturalisation) are discussed in more detail in the following sections.

**Table 3: Matrix of challenges when developing a perceptual model of the intercatchment groundwater flow environment for the purposes of improved hydrological modelling, split by challenge category.**

| Category | Challenge |
|---|---|
| **Data** | |
| **Availability** | Groundwater level data and human influences data can be difficult to obtain and work with, and can also be subject to stringent licensing regulations, including limiting options for publication. |
| **Spatial resolution** | Observations are rarely spatially regular, often "focussed in areas of particular interest" (Barthel and Banzhaf, 2016). Groundwater-related data locations may often not correspond to where a hydrologist might have particular interest. The necessary reliance on historical catchment outlet flow data when undertaking high-level investigations can make identifying IGF particularly difficult (Frisbee et al., 2016). River gauging stations are often not located in optimum locations and so not necessarily measuring all the water exiting a catchment. "Underflow" going below the gauge and/or deeper IGF can be missed (Fan, 2019). |
| **Inconsistency** | Data heterogeneity is a challenge for analysis (Barthel and Banzhaf, 2016), particularly at the regional scale (Refsgaard et al., 2010). Data is collected by a range of different users with consequentially varying quality and/or temporal resolution (Barthel and Banzhaf, 2016; McMillan et al., 2016). |
| **Uncertainty** | Wide range of data uncertainties, including precipitation measurement, PET over/under estimation, the estimation of actual evapotranspiration, gauging station uncertainties, surface water and groundwater catchment boundary locations and human influences data relatability. |
| **Methodological challenges** | |
| **Water balance** | Full consideration of sources of errors (see "Uncertainty" above) should be undertaken, before concluding any water balance non-conservatisms are a result of unmeasured groundwater processes, as highlighted by the Perceptual Roadmap in Fig. 2. |
| **Intercatchment groundwater flow** | To understand how significant it is as a process in a catchment, a targeted review is required (Fan, 2019). IGF processes are challenging to identify, locate and characterise (Frisbee et al., 2016). It can be difficult to quantify (rather than simply perceptualise) IGF indirectly via other variables (i.e. meteorological and hydrological) due to their uncertainties and potential errors. |
| **Groundwater data** | The selection of groundwater data sets can be challenging and require an understanding of the importance of site selection (valley bottom vs interfluve with regards to aquifer properties), particular in hydrogeologically heterogeneous lithologies. The selection of the temporal characteristics of the data also requires careful consideration. |
| **Human influences** | The selection (and application) of appropriate naturalisation methodologies can be difficult and depend on data availability, time and resources, as well as the dominant human influence 'type' (be it point or diffuse impacts, ground or surface). |
| **Perceptual understanding** | |
| **Hydrogeological interpretation** | Local geological knowledge is required to detect and quantify IGF (Fan, 2019), requiring a level of hydrogeological skill/expertise. A limitation of high-level hydrogeological classification is that we are |



| | |
|---|---|
| | consequently not looking at within-unit heterogeneity (Le Moine et al., 2007) (which can be considerable (e.g. in the Chalk in this study (Bloomfield et al., 2011; Marchant and Bloomfield, 2018)), but rather assessing hydrogeological heterogeneity between surface water reaches. |
| **Scope** | It is important to focus on understanding improvements needed for conceptual hydrological models for predictions of now and in the future, rather than a deep understanding of detailed groundwater processes and localised complexity. The scope of the investigation does not, therefore, need to be particularly wide in terms of a hydrogeological analysis. |
| **Human influences** | Abstractions and discharges become particularly important at the regional scale (Barthel and Banzhaf, 2016) but we have also shown them to be key sources of potential uncertainties at the local, reach scale. Their consideration is fundamental to the appropriate identification of IGF. |


### 6.3.1 Input data uncertainties

As summarised in Table 3, the perceptualisation of IGF introduces a wide range of data uncertainties and care needs to be taken to ensure that data anomalies are not mistakenly attributed to IGF (as Bouaziz et al. (2018) and Zanon et al. (2014) warn in the case of input data uncertainties). Whilst a full quantification of uncertainties may be beyond the scope of an IGF
perceptualisation when for the purposes of informing model editing, it is important to reflect on the uncertainties being introduced.

In the case of catchment areal precipitation and PET estimation, the error source is two-fold: sample measurement errors and spot-data interpolation method errors (McMillan et al., 2012). When using a gridded rainfall data product such as CEH-GEAR
(which has undergone extensive raw data and interpolation quality control procedures (Tanguy et al., 2019)), the largest errors associated with the data originate from the raw rain gauge measurements, in particular relating to wind-loss resulting in under-catch (5-16% (McMillan et al., 2012; Keller et al., 2015)). However, the magnitude of these errors is reduced when there is a high rain gauge network density (Keller et al., 2015), such as in the data-rich Thames catchment.

The uncertainties surrounding potential evaporation are complex due to its basis on underlying empirical equations involving a number of separately recorded (or estimated) variables. The potential errors associated with the FAO Penman-Monteith method (on which the CHESS-PE dataset is based (Tanguy et al., 2019)) have been reviewed by a number of authors and shown to vary through the calendar year due to its sensitivity to potential errors in the temperature variable (Talebmorad et al., 2020). Estimates of average potential evaporation error using the FAO Penman-Monteith method have been recorded as 10%
(Talebmorad et al., 2020), 5-10% (Hua et al., 2020) and 10-40% (Westerhoff, 2015).

Discharge uncertainties from river gauging stations vary depending on measurement method (McMillan et al., 2012) and can vary over time (Coxon et al. 2015). Assessment of their errors is complicated by often lacking metadata on the structure and/or



stage to discharge calculation methods (Coxon et al., 2015). In the Thames, weir calculations are most prevalent (Coxon et al.,
2015). Errors in mean time-averaged daily flow measurements can range from 10% to 20% (McMillan et al., 2012). Assessing
discharge over a long time period and incorporating periods of varying climatic conditions can help increase confidence.

Given the above, we believe that average non-conservative reach water balances greater than 100 mm yr$^{-1}$ (from Eq. 3)
(equivalent to 14% of the annual average rainfall in the Thames at Kingston catchment (Marsh and Hannaford, 2008)) can be
assumed to be attributable to factors other than input data uncertainties. However, future work should concentrate on
quantifying these uncertainties in space and time so they can be incorporated in such an analysis.

### 6.3.2 Naturalisation

A notable challenge that we have found in this study is that of the influence of human abstractions and discharges. The process
of river flow naturalisation is itself one of great uncertainty (Terrier et al., 2020). In Fig. 5(b), both >100 mm yr$^{-1}$ water balance
losses and gains have been calculated in Non-aquifer reaches, most notably the three Lower Thames reaches and along the
River Mole (ref. Fig. 1(b)). Following the perceptualisation roadmap in Fig. 2 regarding the absence of aquifer outcrop
coverage and springlines, we can deduce that these "non-conservatisms" are not due to IGF processes and investigate
alternative causes. It is known that major public water supply abstractions (Lower Thames) and discharges (River Mole) are
located in these reaches. The relative scale of these artificial influences in relation to the increase/decrease in river flow between
gauging stations is leading to anomalous water balance results. This highlights a disadvantage and challenge of using reach-
based analysis when applying naturalisation.

In addition, our naturalisation method explicitly discounts groundwater abstractions. In a study area with known abstractions
from nationally important aquifers (Butler et al., 2012), this assumption is obviously flawed. However, it must be
acknowledged that a full naturalisation incorporating both surface and groundwater abstractions and discharges would be a
significantly time-consuming and data intensive programme. Quantifying the impacts of groundwater abstractions on river
flows is challenging (Ivkovic et al., 2014) and more complex than that of surface water influences when, in particular,
identifying the location of any consequential reduction/modification of flow (Coxon et al., 2020).

What our analysis has shown is that human influences from surface and groundwater abstractions/discharges can both mask,
and indeed be misinterpreted for, IGF processes. It is imperative that river flow naturalisation is specifically addressed, and
the impact of any methodological assumptions considered in light of the results obtained.





## 7 Conclusions

In this paper we have shown the prevalence of non-conservative river reaches across the study area, with heterogeneity both
between, and within, geological units giving rise to a complex distribution of recharge and discharge points along the river
network. We have identified likely intercatchment groundwater flow locations and characteristics via a process of data analysis
and perceptualisation, providing an evidence-led challenge to the 'watertight substratum' foundation of many existing
catchment rainfall-runoff models (Le Moine et al., 2007). Outcrops of carbonate fractured aquifers (Chalk and Jurassic
Limestone) show evidence of intercatchment groundwater flow both from headwater to downstream reaches, and out-of-
catchment via spring lines. We found temporal as well as spatial variability across the study area, with more seasonality and
variability in river catchments on Jurassic Limestone outcrops compared to Chalk and Lower Greensand outcrops. Our results
demonstrate the need for local investigation and hydrogeological perceptualisation within regional analysis, which we have
shown to be achievable given relatively simple geological interpretation and data requirements. A lack of representation of
regionally connected groundwater dynamics within conceptual rainfall-runoff models can contribute to core discrepancies in
the annual water balance of a catchment (Pellicer-Martinez and Martinez-Paz, 2014; Fan, 2019) and unrealistic simulations of
observed hydrograph responses (Lane et al., 2019). A model should be improved so that it better represents reality, rather than
rejecting a catchment because of its poor results (Le Moine et al., 2007), but a key challenge is how to link hydrogeological
processes to a surface water-designed existing model, when groundwater science has very different data availability and spatial
and temporal scales. The difficulties in characterising intercatchment groundwater flow have long led to modellers omitting
its representation in models (Fan, 2019). There is a clear need for better evidence-based perceptualisation of groundwater
systems by hydrologists prior to any model edits. Applying a hydrogeologist's thinking to a surface water hydrologist's
'problem', will help to break down the existing 'artificial boundaries' between the two sciences (Fan, 2019; Staudinger et al.,
2019).

*Data availability:* The CEH-GEAR precipitation dataset and CHESS-PE potential evapotranspiration dataset are freely
available from CEH's Environmental Information Data Centre and can be accessed through https://doi.org/10.5285/ 5dc179dc-
f692-49ba-9326-a6893a503f6e (Tanguy et al., 2014) and https://doi.org/10.5285/8baf805d-39ce-4dac-b224-c926ada353b7
(Robinson et al., 2015a) respectively. The recorded/observed discharge datasets used in this study are freely available from
the National River Flow Archive website https://nrfa.ceh.ac.uk/data/search. The human influences (surface water abstractions
and discharges) dataset was obtained from the Environment Agency. The naturalised discharge datasets developed in this paper
unfortunately cannot be made open access due to license restrictions on the human influences data used in the naturalisation
process. Groundwater level data was provided by the Environment Agency.

*Author contribution.:* LDO, JF and GC designed the analysis and LDO carried it out. LDO prepared the manuscript with
contributions from all co-authors.





*Competing interests.* The authors declare that they have no conflict of interest.

*Disclaimer.* Any reference to specific products is for informational purposes and does not represent a product endorsement.


*Acknowledgements.* Louisa Oldham is supported by a studentship funded by the U.K. Research Institute. Part of Jim Freer's time was supported by the Global Water Futures programme, University of Saskatchewan. Gemma Coxon was partially supported by a UKRI Future Leaders Fellowship [MR/V022857/1]. Christopher Jackson and John Bloomfield publish with the permission of the executive director of the British Geological Survey (NERC/UKRI).

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
