# Peer review of "Evidence-based requirements for perceptualising intercatchment groundwater flow in hydrological models"

_EGUsphere, 2022_

## Author Comment (AC1)

**Review of Oldham et al: Evidence-based requirements for perceptualising intercatchment groundwater flow in hydrological models.**

| | RC1: 'Comment on egusphere-2022-529', Anonymous Referee #1, 02 Oct 2022  reply | |
|---|---|---|
| 1. | The study uses an extensive dataset from the Thames catchment to find proof for (variations in) intercatchment groundwater flow (IGF). IGF is hard to quantify, but it is an important process to consider in hydrological modelling. The paper is very well written and well structured. A significant amount of data collection and processing work was done to enable a useful analysis of IGF at this scale. | The authors would like to thank the reviewer for their detailed and constructive comments on our paper.

In response: |
| 2. | On several places in the paper it is stated that both spatial and temporal variations of IGF were studied (e.g. L 19, 112, 605). The spatial variations of IGF are indeed well analyzed. I did not see much about temporal variations of IGF. There is Figure 7 with seasonal patterns in groundwater levels and water balance metrics, which are shortly described in chapter 5.2 and in L476-490. However, the link between this seasonality and the temporal variability in IGF was not described. In addition, there is probably more than just seasonality: what about year to year variations in IGF losses and gains? Do these year to year variations match between losing and gaining stretches (or is there a delay)? These temporal aspects could be either better covered or left out of this paper. | We acknowledge that the temporal analysis presented in the paper is limited to a short description of seasonal patterns in climate, groundwater level, runoff coefficient and water balance and does not report inter-annual variation in IGF. To address your comments, we have removed references to our claim of having studied both spatial *and temporal* variation of IGF. We have edited temporal analysis text (3.2.2 and 6.1 in particular). In addition, we have removed the monthly profiles for water balance from Fig 7 owing to the issues surrounding intra-annual analysis of water balance (in particular given the complexities surrounding intra-annual delays in storage). In the interest of paper length and scope, we have not added additional analysis, but rather re-focussed our temporal analysis section and made edits to the text to clarify that we are not quantifying IGF from this temporal analysis, but rather presenting information on the hydrological and hydrogeological characteristics of (and between) the different hydrogeological units that might be of interest and use to the hydrological modeller when considering IGF edits. |
| 3. | Regarding the spatial analysis: would it be possible to connect losing and gaining stretches, compare the IGF fluxes and maybe combine catchments into larger scale conservative catchments? This may be possible for e.g. the Coln, Kennet, Colne and Mole catchments. | We discuss connecting losing-gaining reach in section 5.1 using the results from the River Kennet as an example. We made the research decision to concentrate on the highest resolution scale possible, i.e. the reach length scale. We have, however, reported the water balance for the Thames at Kingston as a whole in Section 5.1. |

| 4. | L27: We found temporal as well as spatial variability of...? See above regarding temporal variability in IGF. | Please see the response to comment 2. |
|---|---|---|
| 5. | L110-112: Consider to rephrase into an objective statement. | To make the statement more objective, we have reworded it to: "Quantifying the water balance using available national meteorological, hydrological, hydrogeological, geological and human influences datasets enabled us to develop a perceptual model of IGF". |
| 6. | L190-198: Groundwater abstractions are not mentioned here. Later, this is covered and discussed. Still, I am curious whether the volumes of groundwater abstraction are significant enough to have impact on your analysis. Maybe some regional numbers are available? | We have added quantification of groundwater (and surface water) abstraction rates at the Thames at Kingston catchment scale to section 2.

In addition, we have further expanded on the challenges associated with naturalised river flows for groundwater abstraction to support our reasoning for not adopting this approach in our methodology (section 6.3.2) |
| 7. | L271: in->is? | Corrected. |
| 8. | L302: is the->is in the? | Corrected. |
| 9. | L338: it could save a lot of space to only show the naturalized results in fig 4,5,6. The difference is indeed not that large. | Agreed. We have moved all non-naturalised results to a new Supplementary Information document and our figures now only show naturalised results. |
| 10. | L356-357: with the low amount of catchments (4 in LG, 11 in JL) these interquartile ranges are highly uncertain. | We have recognise the small sample sizes and associated uncertainties and have now acknowledged this more explicitly in the text in Section 5.1. |
| 11. | You could also choose to plot the averages with error bars to show the (variable) uncertainty of the statistics. | We feel that the interquartile ranges shown on Figure 4 already illustrate the uncertainty of the statistics we have presented. |

| 12. | Figure 5: the catchment boundaries are unclear in these maps. Would more legend colors be possible? | Legend colours have been changed to improve clarity for the reader. Thank you for this comment, we believe the Figure is improved. |

---

## Author Comment (AC2)

**Review of Oldham et al: Evidence-based requirements for perceptualising intercatchment groundwater flow in hydrological models.**

| | RC2: 'Comment on egusphere-2022-529', Anonymous Referee #2, 10 Oct 2022  reply | |
|---|---|---|
| 1. | Non-conservative reaches and catchments are popular in groundwater-dominated regions and karst areas. Perceptualizing of hydrological processes in these regions is of great importance as it enables us to recognise where intercatchment groundwater flow (IGF) may be occurring and highlights the need for local investigation. In this study, a framework is proposed to evaluate the spatial and temporal IGF and applied to the River Thames with wealth of data and densely gauged river network. It is an interesting topic in hydrology, and the manuscript is well organized. However, there are still several problems and deficiencies in the paper and further revision is needed. | The authors would like to thank the reviewer for their detailed and constructive comments on our paper.

In response: |
| 2. | The water balance is the basic metric to recognize the IGF and the term AET plays a key factor in determine the metric. However, the estimation of the AET contains great uncertainty and AET has great spatial variability in large mountainous basin. The uncertainty in estimate of AET and then water balance metric should be analyzed. | For the purposes of quantifying why we can state that our interpretation of IGF processes via water balance (WB) anomalies are justifiable, rather than just uncertain, we will present an uncertainty evaluation of not just AET but the P, Q and AET time series estimates. This will use a simple error model to generate multiple time series for an example catchment as per Lloyd et al. 2016 and calculate the resultant WB uncertainty range. From this we will be able to state more categorically why our thresholds for considering a WB anomalous, and thus attributed to IGF type processes, can be stated.

Lloyd, C.E.M., et al., Discharge and nutrient uncertainty: implications for nutrient flux estimation in small streams. Hydrological Processes, 2016. 30(1): p. 135-152. DOI: 10.1002/hyp.10574 |
| 3. | Line 393-410. The analysis of water balance at inter-annual scale should be careful, as the temporal variation of water balance metrics is more complex than that for the multi-year average condition. For example, the soil water storage is a nonnegligible term for water balance. Further, the change of groundwater level is mainly controlled by local hydrogeological conditions. That's for sure, there are significant differences in the temporal variation of hydrological factors among | Firstly to say we are answering this reviewer comment by assuming they meant intra-annual not inter-annual. We agree that intra-annual analysis of water balance is indeed challenging but we would argue that one can still explore seasonal patterns of behaviour if the assumptions are recognised when doing so. However, we have re-focussed our temporal analysis sections into a review of the climatic and river flow components of the water balance, and groundwater levels. We made edits to the text to clarify that we are not |

| | | |
|---|---|---|
| | hydrogeological units. But I don't catch that how these reflect or indicate the differences in IGF. | quantifying metrics of IGF from this analysis, as we do not believe that is possible from a reach-length based water balance analysis. However, we are inferring that by presenting information on the hydrological and hydrogeological characteristics of (and between) the different hydrogeological units that these are evidence of the fact that IGF flows might be important and such analysis has value for the hydrological modeller when considering where IGF processes may be needed. |
| 4. | The perceptual model of the Thames is of great importance in the paper (Figure 8). But it seems confusing as too many lines and explanatory text. I suggest authors reorganize the figure 8. | We purposefully designed a perceptual model diagram that incorporated the visual mapping and qualitative information elements common in hydrogeological conceptual models, to illustrate the approach commonly taken by hydrogeologists and the method of presenting a wealth of information in one figure. We admit that the figure is busy but think that reducing or reorganising it would reduce its impact. |
| 5. | A description of climate, especially the spatial and temporal variability of P and AET, is needed for the basin in the section of study area. A brief introduce of the runoff depth and its temporal varation for the basin is also needed. | We have added text on rainfall, PET and runoff in the Study Area section and added a figure in our new Supplementary Information document showing their spatial variability across the catchments. AET is one of the products of our analysis, as is the investigation of temporal variability, so we feel that these elements should be introduced later than in this Study Area section. |
| 6. | It will be helpful to understand the degree of losing and gaining of reaches. | We agree that this is of high interest given the subject topic of IGF, but again is a product of our results and therefore we feel it appropriate to only be discussed in the Results section. |
| 7. | Line 220. "A positive residual" is , which should be pointed out clearly. | Done. |
| 8. | Line 265. More explanatory text is needed for figure 2. | We have developed the text to strengthen the link between the literature quoted and the figure (section 3.2.3), and further explained the use of the perceptual roadmap in the identification of IGF. |
| 9. | Line 380. In figure 5, the water balance metric is greater than 1000 mm/yr in several catchments. The value seems too large for the region. The authors may check it carefully. | The unusual water balance results stem from the combination of a number of different factors, all highlighting the challenges when undertaking such an analysis. Firstly, we are calculating water balance at the reach, not catchment, scale. Significant differences between the topographical surface water catchment and the underlying groundwater catchment are |

| | | exacerbated when discretising datasets based on topographic boundaries. In addition, the uncertainties associated with the location of, and scale of, human influences are considerable when assigning reach-scale impacts. We discuss how the >1000 mm/yr results in the Lower Thames are likely as a result of these surface water abstraction and discharges in section 5.1, directing the reader to the more detailed discussion on the topic of naturalisation in section 6.3. |
|---|---|---|
| 10. | Line 413. In figure 7, what are the means of shadows in the sub-figures (a)-(g) and different colors of curves in (h)-(o). | An explanation of the shadows and different coloured lines was indeed missing and has been added to the figure caption. Thank you for raising this omission. |
| 11. | Lines 600-6015. In this section, the authors should focus on what you have found in the paper rather than suggestions. | The authors feel that a combination of both a summary of what we have found and suggestions for further work are of merit in this section, in particular as our subsequent papers will be aiming to address some of the issues and "further work" topics we make note of. |

---

## Author Comment (AC3)

**Review of Oldham et al: Evidence-based requirements for perceptualising intercatchment groundwater flow in hydrological models.**

| | | |
|---|---|---|
| **RC3: 'Comment on egusphere-2022-529', Anonymous Referee #3, 11 Oct 2022  reply** | | |
| 1. | It is a common fact that water actively exchanges between surface divides in regions with carbonate fractured aquifer outcrops. This have challenged the 'watertight substratum' assumption that is the foundation of many existing catchment rainfall-runoff models for long time without appropriate solutions and model conceptulizations. The main aim of this manuscript is to improve our perceptual models of intercatchment groundwater flow. The authors took advantage of their wealth of data, densely gauged river network, and geological variability from national meteorological, hydrological, hydrogeological, geological and artificial influence datasets to develop a perceptual model of intercatchment groundwater flow (IGF) and to show how it varies spatially and temporally in 80 subcatchments of the River Thames, United Kingdom (UK). The water balance, presence of gaining/losing river reaches and intra-annual dynamics were investigated through a water balance analysis. | The authors would like to thank the reviewer for their detailed and constructive comments on our paper.

In response: |
| 2. | The study is important for hydrological predictions and water resources management in groundwater-activated catchments. However, the method adopted by the authors can only provide site-specific results about qualitative water balance, it is still difficult to represent regional inter-catchment groundwater dynamics as they could not provide some essential functions that describing how groundwater between neighbor units exchanges according to different conditions of groundwater levels, different lithology, human abstractions and so on. In order to couple IGF processes into existing hydrologic models, it is important for the authors to derive the IGF functions quantitatively describing how IGF varies with time, groundwater levels and abstractions, …. | The reviewer has made an important point, however we would argue that we provide a broad summary of these influences in quantifiable terms, rather than being wholly qualitative. We have not dealt with the detail, as that is beyond the inference possible with the methodological approach we have taken. Our overall aim was to identify if water balances at the reach scale are anomalous, not by how much in specific directions of flow, as we cannot make that inference from the information we have. We wanted to produce a high-level perceptual model that could be used by a hydrologist to focus model development flexibly, rather than provide absolute threshold limits for subsurface fluxes. In addition, our human influences data is highly uncertain and we did not feel comfortable reporting exact values of IGF derived from them – water balance analysis can only take us so far.

We have made edits to the introduction and analysis sections in order to clearly state our aims and objectives regarding identifying (and not explicitly quantifying) IGF, in order to better manage the expectations of the reader |

| | | |
|---|---|---|
| | | and justify our methodological approach, including the limitations of defining exact amounts. |
| 3. | The water balance equations (1)-(4) adopted are also not rigorous as discussed by the authors themselves in Section 6.3 that input data uncertainties can lead to large computational uncertainty. In fact, equations (1) or (3) represent multiyear water balance instead of single year water balance. So dS/dt=0 is not strictly true, and a empirical 100 mm/yr was used by the authors to help to identify the non-conservative reach water balance. As the IGF fluxes could not be measured directly in catchment scale, empirical estimation is inevitable. However, the fudge factors e.g., 100 mm/yr as well as the physical meaning of S (groundwater storage or soil water storage?) should be discussed in depth. | As per the wider IGF literature (e.g. Le Moine 2007, Bouaziz 2018, Fan 2019), S has been used by us as a general term to incorporate all storage in a catchment, i.e. groundwater storage, soil storage, vegetation storage etc. Over the long-term the change in storage in a catchment can indeed be considered to be negligible. For this reason we have not tried to quantity IGF at the intra-annual scale. Our water balance calculations based on equations (1) and (3) are indeed at the annual scale and have been used by many other authors. We have added some text on the limitations of assuming the change in storage to be negligible at the annual timescale in the Methods section.

Regarding the uncertainties in input data, we will present an uncertainty evaluation of the P, Q and AET time series estimates. This will use a simple error model to generate multiple time series for an example catchment as per Lloyd et al. 2016 and calculate the resultant water balance uncertainty range. From this we will be able to state more categorically why our thresholds (i.e. 100 mm/yr) for considering a water balance anomalous, and thus attributed to IGF type processes, can be stated. We have also included further discussion on our decision-making process regarding the threshold value in the discussion section.

Bouaziz, L., Weerts, A., Schellekens, J., Sprokkereef, E., Stam, J., Savenije, H. and Hrachowitz, M.: Redressing the balance: quantifying net intercatchment groundwater flows, Hydrol Earth Syst Sc, 22, 6415-6434, https://doi.org/10.5194/hess-22-6415-2018, 2018.
Fan, Y.: Are catchments leaky?, Wires Water, 6, https://doi.org/10.1002/wat2.1386, 2019.
Le Moine, N., Andreassian, V., Perrin, C. and Michel, C.: How can rainfall-runoff models handle intercatchment groundwater flows? Theoretical study based on 1040 French catchments, Water Resour Res, 43, https://doi.org/10.1029/2006wr005608
Lloyd, C.E.M., et al., Discharge and nutrient uncertainty: implications for nutrient flux estimation in small streams. Hydrological Processes, 2016. 30(1): p. 135-152. DOI: 10.1002/hyp.10574 |

| | | |
|---|---|---|
| | | |
| 4. | The quality of many figures could be further improved e.g., to fully show the reach units are subdivided and to accompany their figures tightly with the text words to upgrade the readability. The reviewer suggests that reach units can be subdivided into two categories, the headwater reach and the internal reach. The water balance of reach units from the headwater areas which is recharged singly by precipitation in conservative catchments should be highlighted in order to identify the leakage recharge from outside catchment. | Reaches have been subdivided into headwater and non-headwater (internal) reaches in Fig 6 and its accompanying text. We also reflect on the importance of whether a reach is a headwater or not in Fig 2 and its accompanying text. |
| 5. | P4L120: Here annual average precipitation for the whole basin and its spatiotemporal distribution is needed. Discharge volume of main gauges also should be provided. | We have added text on catchment rainfall, PET and discharge in the Study Area section and added a figure in our new Supplementary Information document showing the spatial variation in catchment P, PET and discharge. |
| 6. | P6L144-146: "reach as the catchment area between river gauging stations. The analysis undertaken in this study is developed at the river reach scale rather than at the sub-catchment scale". However, the title of this manuscript is "…perceptualizing inter-catchment groundwater flow…". What is the difference between reach drainage area and sub-catchment? | We have defined a reach as "the catchment area between river gauging stations". The drainage area of a sub-catchment is therefore much larger – incorporating all upstream reaches. We acknowledge that we use the term "inter*catchment* groundwater flow" rather than "inter*reach*", but feel that we should continue using the well known term despite this discrepancy. |
| 7. | Are the units presented in Figure 1c the reach units? I suggest that the authors provide reach units distribution map in terms of the river gauging stations. | The units presented in Figure 1 (and also now Figure S1) are catchment units, not reach units. We have edited the figure caption to stress this.

Reach calculations are a part of our analysis. We have used Figure 1 to show catchment information and data that is freely available from external sources to provide a general background to the study area. We feel that the separation between catchment data in the general background section and reach data in the results and discussions sections mirrors the development of our analysis. |
| 8. | P7L175: Provide the cells adopted for water balance computations. | On the assumption that the reviewer is referring to "wells" rather than "cells": |

| | | We show the location, distribution and average groundwater level of the 151 wells that we use to develop our monthly groundwater level profiles and support our groundwater flow direction reviews in Figure 3a. |
|---|---|---|
| 9. | P8L175: "A limiting factor of 70% of the total reach area was assigned as an indicator of reach coverage." What is the meaning of 70% here. | We have reworded the sentence to improve clarity for the reader. |
| 10. | P8L217-219: S represents many storage components, e.g., groundwater storage, soil water storage, vegetation water storage, etc. how do they calculate groundwater exchanges without eliminations of other terms. It is possibly due to this reason I guess that an empirical factor 100mm/yr was adopted (see in Lines 573-576), which helps to filter disturbance from other terms? In addition, equation (1) or (3) can represent multiyear water balance instead of single year water balance. So the authors should explain the limits of using these equations. | We have added text to the Uncertainties section in the Discussion to explicitly reference the limits of using equations 1 and 3 and assuming the change in storage to be negligible.

See our response to comment #2 regarding the selection of the 100 mm/yr empirical factor. |
| 11. | P12L315-316: "Due to the high storage (Table 1)". In table 1 lower greensand aquifers are with the lowest average (0.005) storage coefficients? Why you claimed the high storage in the main text? Similar expressions can be seen also in P11L290, P21L479. | We have made edits to the text in Section 4 to ensure our descriptions regarding hydrogeological physical properties of the aquifers are valid. This includes corrections to descriptions of storage. |
| 12. | P13L340-341: "The three lowest main river reaches show particularly large naturalised water balance losses (>1000 mm yr-1)". I noticed that the average annual precipitation of Thames basin is only about 710 mm (Gabriel et al., 2022). Why so much losses of water (>1000 mm yr-1) in the main reaches in the River Thames? | The unusual water balance results stem from the combination of a number of different factors, all highlighting the challenges when undertaking such an analysis. Firstly, we are calculating water balance at the reach, not catchment, scale. Significant differences between the topographical surface water catchment and the underlying groundwater catchment are exacerbated when discretising datasets based on topographic boundaries. In addition, the uncertainties associated with the location of, and scale of, human influences are considerable when assigning reach-scale impacts. We discuss how the >1000 mm/yr results in the Lower Thames are likely as a result of surface water abstraction and discharges in section 5.1, directing the reader to the more detailed discussion on the topic of naturalisation in section 6.3. |

| | | |
|---|---|---|
| 13. | Do you have the losses averaged over the reach units? In P14L349-357, other values about water losses or gains seems to be regular. However, I don't understand how do you convert the water losses into water depth. I suggest that the authors may use water losses volume in m3 yr-1 instead of water depth since the reference reach unit area is quite difference and upstream inflow is also different from up to down river reaches. | The water balance variables of reach P, reach AET and reach Q are averaged over the reach area to obtain values in mm/year. We purposefully chose to report our water balance analysis in depth (mm/year) rather than volume for both consistency across different climatological and hydrological variables and to account for the differences in reach areas. We feel that this remains the best choice. |
| 14. | P14L362: what is the raio of 622 mm yr-1 annual loss in total volume of precipitation in the Kennet headwater reaches. As we know, headwater reaches do not receive upland surface inflow, so the net loss of 622 mm is large compared to the annual precipitation 710 mm over the whole basin. | Some of the non-conservatisms we find on aquifer outcrop areas can be significant, and highlight the impact that regional groundwater systems (i.e. those operating across reach topographical catchments) can have on river reach water balances. However, 622 mm/yr was a sum of all the Kennet headwater reaches' non-conservatisms. 710 mm/yr is a whole Kennet catchment average for rainfall, so the two values are not comparable. The average non-conservatism of the Chalk headwater reaches in the Kennet is 125 mm/yr and, in comparison to the average rainfall in those reaches, is actually 15% of precipitation. We have amended the text to report average losses and added reference to its ratio to the average precipitation. The use of a cumulative mm/yr headwater reaches loss was erroneous and we thank the reviewer for drawing this to our attention. |
| 15. | P19: I suggest that the total amount of groundwater exchange should be marked in Figure 8. And how do you judge the flux directions? From the method in Section 3, I do not find related algorithm for estimating the flux directions. | As we discuss in our response to comment number 2, we made the research decision not to explicitly quantify IGF as our overall aim was to identify where water is moving that is not controlled by topography, not by how much. In 3.1.2 we state that our groundwater level data were used to confirm groundwater flow directions We have added text to Section 3 to further explain our method for estimating flux directions, whereby it is based on comparison of water balance results against our groundwater level data and published groundwater level contours. |
| 16. | P20L454-455. "The Chalk of the Thames Basin can be locally sub-karstic, but fracture and fissure flow remain the primary groundwater flow". It maybe true as you claimed, however, if the IGFs should occur in the relatively less passageways of karstic conduits? | There is one case in the literature of IGF between topographical catchments occurring in the Chalk of the Thames via sub-karstic flow (see Maurice et al. 2022) but fracture flow is by far the dominant flow process. |

| 17. | P21L503. It is true that not including IGF as a model flux will result in many models overestimating river flows or actual evapotranspiration. But the key question may be to describe how groundwater between neighbor units exchanges according to different conditions of groundwater levels, different lithology, human abstractions and so on. | This would indeed be interesting, and necessary, further work but at this stage we feel this is outside of our scope. |

---

## Author Response (AR1)

Dear Dr de Rooij,

Thank you for your review of our reviewer responses. We have actioned our proposed changes/edits as per our outlined reviewer responses. With regards to your specific points:

- We have edited paragraph 3 of Section 1.3 in the Introduction to add preceding and subsequent sentences that clearly state the objectives of the study: "*This paper presents a high-level perceptualisation of inferred IGF for the purposes of rationalising the spatial development of catchment based hydrological conceptual modelling.* **Quantifying the water balance using available national meteorological, hydrological, hydrogeological, geological and human influences datasets enabled us to develop a perceptual model of IGF**, *where we have identified the location and direction of IGF at a high-level. Our perceptual model does not quantify the detailed specifics and directions of IGF, owing to the underlying information and analyses we have available. However, we can identify river reach lengths that have anomalous water balances that infer IGF processes are likely to be needed in the development of catchment based geospatial modelling environments, which is our core aim with this research*."

- Our new quantitative estimates of water balance errors support our qualitative analysis and methodological choices. Our ratio of qualitative to quantitative analysis reflects our overall aims and objectives of the study, whereby we infer where IGF processes are likely to be needed in the development of catchment based geospatial modelling environments. We have made these aims clearer throughout the paper in response to the reviewer comments.

- We feel that Intercatchment Groundwater Flow is an appropriate term to use to describe the groundwater water flux beneath topographic boundaries. The term has been used to refer to inter-**reach** fluxes in numerous similar papers (e.g. Le Moine et al., 2007; Bouaziz et al., 2018; Fan, 2019) and we feel its use would help our paper integrate with the existing literature.

We trust that the edits we have made address the comments and suggestions from you and our three reviewers. We look forward to hearing your response.

Regards,

Louisa Oldham

---

## Author Response (AR2)

Dear Dr de Rooij,

Thank you for your recent review and acceptance of our manuscript subject to the three minor changes you outlined. In response:

- I have moved the BFI definition to the acronym's first use. Apologies for the oversight.
- I have removed the extra full stop.
- The sentence is meant to read "with flexibility a key principle", rather than "as".

Regards,

Louisa Oldham